# Dynamic Bilingualism to Dynamic Writing: Using Translanguaging Strategies and Tools

Onudeah D. Nicolarakis * and Thomas Mitchell

School of Language, Education, and Culture, Gallaudet University, 800 Florida Avenue NE, Washington, DC 20002, USA
* Correspondence: onudeah.nicolarakis@gallaudet.edu

**Abstract:** This study is a qualitative analysis of a naturally occurring translanguaging phenomenon in the writing practices of fifteen high-scoring deaf bilingual adult writers. This study aims to identify translanguaging factors related to writing achievement and explore themes that emerge within an asset-based/antideficit, deaf bilingualism/Deaf Gain theoretical framework. Data were gathered by collecting, reviewing, coding, and identifying overarching themes in the interview transcripts. The findings show that high-scoring deaf participants utilized translanguaging writing strategies and tools such as *translingual interdependence*, *language flexibility*, *semiotics and multimodalities*, *American Sign Language (ASL) drafting*, and *visual tracking skills* that led to their writing achievement.

**Keywords:** translanguaging; dynamic bilingualism; writing achievement; deaf adults; asset-based/antideficit; Deaf Gain

## 1. Introduction

In the field of bilingual and multilingual literacy, translanguaging is an approach that positively embraces and sustains the language experiences of all individuals and encourages the fluidity and intermingling of their collective languages in writing. However, research specific to the writing strategies and tools that leverage the collective linguistic and semiotic repertoires of deaf bilingual and multilingual learners is still in its infancy. The terms bilingual and multilingual have unique applications to deaf learners where the term bilingual refers to deaf individuals who use two languages (e.g., a signed and a spoken language or two signed or two spoken languages). Deaf multilingual learners (DMLs) generally refer to deaf learners whose home language is other than English and those who may use two or more languages (Cannon et al. 2016; Pizzo 2016). Therefore, bi-/multilingualism will refer to both groups of deaf learners. Furthermore, translanguaging does not suggest American Sign Language (ASL) or English as the default language for deaf learners. Translanguaging applies to all deaf individuals, including those who use multiple languages other than ASL or English, such as Mexican Sign Language (LSM) and Spanish. In this paper, the term deaf refers to a broad spectrum of deaf learners with all degrees of hearing levels, abilities, language preferences, and signing modalities.

Translanguaging is a range of flexible and dynamic semiotic and linguistic practices that bi-/multilingual individuals access through various languages and modalities to serve communicative and language purposes, including writing (Blackledge and Creese 2017; Canagarajah 2011; García 2009, 2012; Hornberger 2016; Wei 2017). When translanguaging, bi-/multilingual individuals leverage their available multilinguistic resources to amplify their linguistic competencies, even if the output is in just one language (García and Wei 2014; Zapata and Laman 2016). One common misconception about bi-/multilingual individuals is that they are two "monolinguals in one person" (Grosjean 1989, p. 3). Humans construct social boundaries of named languages (e.g., English, Spanish, or Tagalog) and draw orderly distinctions between these language borders, but the language centers of our brains do not recognize these distinctions (Petitto 2009). A single, unified linguistic system in the brain

disregards those borders of named languages and utilizes all linguistic resources, including languages that are not actively used, as a unified entity (García and Wei 2014). Therefore, as bi-/multilingual individuals move between and across their collective linguistic repertoire, creating boundless language productions, they are engaged in translanguaging practices (Cenoz and Gorter 2020; García 2012). The use of translanguaging does not hinder the collective language competencies of the individual; instead, they can transcend standardized and prescriptive language practices (e.g., monolingualism) to facilitate more creative and interpretive forms of language (Blackledge and Creese 2017; Otheguy et al. 2015).

In the school environment, studies have shown how translanguaging is a critical multimodal language tool that provides much-needed scaffolding (Vogel and García 2017) to support content learning opportunities (Hoffman et al. 2017) and develop metacognitive skills (Makalela 2015). Furthermore, translanguaging fosters student agency (Blackledge and Creese 2017) and self-regulation (Velasco and García 2014). In schools and beyond, translanguaging extends marginalized learners and their communities alternative pathways to bring their sociocultural, linguistic, and political contexts to social and educational settings (García and Leiva 2014). Due to the growing research on translanguaging and its documented benefits, the focus has shifted to how translanguaging strategies and tools may be incorporated into classroom pedagogy to support complex linguistic tasks such as writing.

## 2. Translanguaging in Writing

Translanguaging can be used as a pedagogical writing tool that is embedded in the writing process to leverage the dynamic linguistic repertoire of diverse writers. Writing is a circular process that involves prewriting, drafting, revising, and editing that requires vast linguistic resources and planning on the individual's part to create written texts. At every step, the stages of writing intersect with language and show that "language is inextricably intertwined with all of these components of the writing process" (Espinosa et al. 2016, p. 1) by incorporating the cognitive, linguistic, and conventional elements of writing (Hammill and Larsen 2009). Thus, writing components dovetail with translanguaging strategies and tools to create a recursive cycle where bi-/multilingual individuals can engage the entirety of their multilinguistic and semiotic resources at various points of the writing process.

Regardless of the many writing models or strategies employed in the classroom, translanguaging pedagogy is a readily available tool that provides targeted support at various phases of the writing process. In the planning stage of writing, students should be encouraged to tap into their entire bi-/multilingual repertoire, such as brainstorming or making notes in any language they can access (Makalela 2015), drawing upon multimodalities such as visual essays, writing from photography, multimedia production writing, and drama (Espinosa et al. 2016), or drawings and videos (Velasco and García 2014 to engage their flexible and dynamic linguistic repertoire. The full range of linguistic repertoire is essential to the drafting process where bi-/multilingual writers enlist problem-solving skills "at the word, sentence, and whole-text levels" (Velasco and García 2014, p. 10) that may include multilingual resources such as dictionaries, using the internet, or comparing multilingual texts (Hesson et al. 2014). Teachers and students work together in the production stage to establish language goals, such as what language and modalities to use to produce their compositions. The use of multimodalities in writing supports students' rhetorical composition development and empowers them to have discourse in their multiple communities (Bohannon 2015).

The intentional use of translanguaging, as part of teaching and learning, generates positive outcomes. Research by Velasco and García (2014) found that translanguaging pedagogy resulted in bi-/multilingual writers who achieved "higher standards of thought, creativity, and language use" (p. 7). For instance, dual-language programs benefit from translanguaging practices that support the writing process when there are intentional translanguaging spaces in the curriculum (Hesson et al. 2014). Standard writing models, such as writer's workshops that embed translanguaging, support students' communicative,

writing, and cultural competencies (Burke and Holbrook 2018). Translanguaging in the bi-/multilingual classroom allowed emergent bilinguals to fully enable their cultural and language resources and shed deficient language labels (García 2009, 2011). For example, in *Teaching English to Speakers of Other Languages* (TESOL), translanguaging reinforced the development and progression of the student's writing skills (Piccardo 2013).

The use of translanguaging is a naturally occurring phenomenon in the bi-/multilingual classroom; however, translanguaging needs to be deployed as a pedagogical tool instead of an intuitive practice (Canagarajah 2011) in various contexts, such as writing instruction. Despite intuitive practices of translanguaging in the classroom and writing, language ideology and the emphasis on monolingualism and the production of English-only text makes it challenging to utilize translanguaging as a planned pedagogical tool (Velasco and García 2014). Vallejo and Dooly (2020) argued that the longstanding tradition of monolingual orientation to writing places the mastery of prescriptive English language practices at the center of writing excellence, deepening the marginalization of bi-/multilingual learners. The monolingual orientation of writing education presupposes a recursive model of language learning, where each language is taught separately and in succession (i.e., English as the first language and the student's home language as a second language). The subtractive approach to the plurality of languages students bring to the classroom deters the bi-/multilingual learners from engaging their existing linguistic resources in other languages for writing success (Cenoz and Gorter 2020; Seltzer and de los Ríos 2021).

Additionally, there is polarization of English from other written languages, as seen with language labels, such as "*nonnative*, *L2*, or *ESL* identify learners according to a single scale of reference—i.e., their relative proficiency in English" (Canagarajah 2015, p. 417). Based on their relative position to English proficiency, these terms result in various narratives of bi-/multilingual writers as deficient, reluctant, or unwilling. Pushed to the periphery are emergent and advanced bi-/multilingual writers who must write in one language, such as English, without structured writing tools such as translanguaging that utilize their collective linguistic system. Translanguaging in writing rejects the recursive view that language and writing are taught in linear succession and instead accepts the translingual view, which is the synergy between multiple languages in the writing process (Canagarajah 2015). Additionally, translanguaging creates spaces for students to use their "full linguistic repertoire without regard for watchful adherence to the socially and politically defined boundaries of named (and usually national and state) languages" (Otheguy et al. 2015, p. 83). Translanguaging spaces leverage the collective linguistic and semiotic repertoires of bi-/multilingual writers and counter the label that bi-/multilingual students are deficient, reluctant, and unwilling writers.

*Deaf Learners: Translanguaging in Writing*

The paucity of research in translanguaging and writing for deaf learners results in a limited understanding of how translanguaging as a pedagogical writing tool may support deaf learners. An important exemplar is how research in translanguaging in deaf communities does not parallel the lived linguistic realities of deaf individuals:

> "A common theme across the signing Deaf experience worldwide is the constant navigation and mediation of at least two languages and the multiple abilities (signing, fingerspelling, reading, writing, mouthing, lip-reading, speaking, gesturing) that are at their disposal, although some are more or less accessible depending on the individual". (Gárate-Estes et al. 2021, p. 234)

In the context of deaf learners and language use, Humphries and Allen (2008) relates the common deaf learning experience to the presence of both spoken and natural signed languages. Similar to bi-/multilingual classrooms, where translanguaging is natural and often intuitive (Canagarajah 2011), those practices are present in the deaf classroom (Gárate 2012; Swanwick 2017). Based on the widespread practice of translanguaging, one can build on the premises offered by Gárate-Estes et al. (2021) that deaf learners are inherent translanguagers tasked with navigating the writing curriculum in a heterogeneous

language environment. However, the prevalence of a monolingual writing orientation heightens scholarly concerns that translanguaging practices in deaf education may perpetuate the misallocation of spoken languages as the monolingual goal of these deaf learners (De Meulder et al. 2019; Snoddon 2017). One prominent example is the use of simultaneous communication (sim-com) as the primary source of instruction. Sim-com is when an individual attempts to simultaneously produce both spoken and manual language modes for each word (Schiavetti et al. 1996). Often when one sim-coms, the output relies on the order of the spoken modality rather than what makes sense conceptually in sign language (Johnson 1983). Translanguaging practices may result in the incidental mixing of two languages, such as chaining a signed concept and connecting it to a spoken language. The use of translanguaging strategies may encompass the occasional mixing of language to support comprehension. However, sim-com is neither a means to translanguaging instruction nor a bona fide language for deaf classroom instruction (Wang et al. 2017). One of the shortfalls of using sim-com as an instructional tool is that it can be sensory-disorienting as both modalities being used simultaneously can cause confusion and a fragmented language experience for deaf students (Swanwick et al. 2022). In the Swanwick et al. (2022) study, deaf students missed key information and meaning making opportunities when their teacher used sim-com to teach new concepts, because they did not have full access to a clear language modality. In contrast, Swanwick (2017) argued that translanguaging in the deaf classroom has been well-received in addressing the entire linguistic repertoire of deaf learners—signed, spoken, and written languages—as long as translanguaging recognizes signed and spoken languages as complete languages. Thus, the further development of befitting translanguaging applications and practices in the deaf classroom that encompasses the plurality of languages and modalities exhibited by deaf learners proffers pathways to translanguaging as a pedagogical writing tool.

Emergent research in the past decade on translanguaging and deaf people offers various glimpses of how these strategies and tools benefit deaf individuals in their communities and schools. For example, a study of deaf bi-/multilingual adults and reading found that translanguaging supported the development of metacognitive skills and bilingual language development (Hoffman et al. 2017). A research study of translanguaging in deaf university professors concluded that they used an expansive semiotic repertoire to utilize multimodal and multilingual resources to create a vibrant, visually oriented presentation of content (Holmström and Schönström 2018). In a different study of deaf bi-/multilingual adults, translanguaging with interpreters was a source of language negotiation to resolve communicative conflicts between hearing and deaf individuals (Napier et al. 2022). In one study on translanguaging in deaf writers, translanguaging entails using translation where deaf bi-/multilingual individuals translate inner American Sign Language (ASL) dialogue into written works (Swanwick 2017).

With the focus of this study on translanguaging and writing, it is necessary to identify previously known translanguaging pedagogical strategies and tools in the literature that may be relevant to positive writing outcomes for the deaf learner. For example, bi-/multilingual individuals may intuitively practice brainstorming or making notes in any language, using multimodalities such as visual essays, writing from photography, multimedia production writing, drama, drawings, and videos in the classroom. Gárate (2012) included signing, viewing, listening and speaking, reading and writing, fingerspelling, chaining, translation, English mouth visemes, and lip-reading as different forms of language domains and modalities essential to translanguaging practices in the classroom. For instance, chaining connects a series of meanings and concepts in more than one language, and modality is used up to six times as much in deaf classrooms compared to hearing classrooms (Humphries and MacDougall 1999). Horner and Tetreault (2017) posited that the general population of bi-/multilingual people engage in translation by negotiating the meaning of what they write between two or more languages. Translation is common in the deaf classroom as teachers and students move between different languages and modalities. Purposeful concurrent language use in deaf classrooms, a translanguaging strategy that

uses multiple signed and spoken language modalities, supports the development of ASL and English (Andrews et al. 2016). Some other standard features of translanguaging in writing include script-switching, postponing, vocabulary revisions, and rehearsing (Velasco and García 2014). It is important to note that scholars are increasingly adopting the view that translanguaging strategies are not the simple use of distinct words, sounds, and morphology from separate languages, but rather "they are simply the bilingual's words, sounds, and morphology that bilinguals learn to then suppress or activate when they are in different communicative situations" (García and Lin 2017, p. 16). This translanguaging view represents the fluidity of translanguaging strategies as bi-/multilingual individuals engage their collective linguistic resources and how (i.e., in what language) they choose to achieve their communicative and language purposes.

Due to the variability in the linguistic repertoire of deaf learners (Gárate 2012), mixed with the intuitive practice of translanguaging (Canagarajah 2011), this study aims to identify the translanguaging strategies that deaf adult bilinguals use to direct their writing experience. García and Wei (2014) argued that researchers must look at the practices of bi-/multilingual writers and the translanguaging strategies they utilize to understand better the role of translanguaging in writing. Additionally, the general literature on bi-/multilingualism and translanguaging has scarce studies on the effects and the roles of multimodalities (Blackledge and Creese 2017), consistent with the literature gaps on translanguaging and multimodalities in bi-/multilingual deaf writers. Therefore, this study aims to investigate the diverse bilingual and multimodal language and writing experiences of bilingual deaf adults to understand better the role of translanguaging strategies in the writing process.

## 3. Theoretical Framework

The theoretical framework of the present study consists of *asset-based/antideficit*, *deaf bilingualism*, *and Deaf Gain critical theory* (Bauman and Murray 2014; García and Cole 2014; Harper 2010, 2012). As in studies that aim to deviate from, disrupt, or challenge the status quo, critical theory was needed to frame and establish the theoretical lens of our research. Critical theory is rooted in the idea that injustice and subjugation shape the lived world (Kincheloe and McLaren 2002). It also challenges the research based on behaviorism and specific epistemological beliefs that result from discrimination being imbued in the perspective of those in positions of power that enable them to define interpretation (Kincheloe and McLaren 2002).

According to Kincheloe and McLaren (2002), many critical theories are constantly evolving and being created based on identifying systems in power and the specific ideology being challenged. For instance, our intention in the present study is to deviate from the perspective on deaf bilinguals based on a deficit model as seen in intervention studies (Easterbrooks and Stoner 2006; Wolbers and Dostal 2010; Wolbers et al. 2015) and to offer in its place a reframed perception focusing on language and biological diversity.

By studying the translanguaging writing strategies and tools that helped skilled adult deaf bilingual writers write, one could observe their perspectives on knowing two languages and modalities. The uncovered information could then contribute to the enrichment of deaf communities and society as a whole (Bauman and Murray 2014; Harper 2010). The perception of making a meaningful contribution to society is referred to as *Deaf Gain*. Instead of a narrative in which deaf people are depicted as a burden on the world, a narrative is proposed that portrays deaf people as having an instrumental role in human development. This reframing shifts the focus from needing to become hearing to having an advantage by being a deaf bilingual. At the same time, it recognizes the language and translanguaging dynamic bilingualism skills that deaf people consistently use in synthesizing the use of both American Sign Language (ASL) and English to produce writing (García and Cole 2014). The decisions and translanguaging strategies derived from their knowledge of languages and modalities, especially determining when to think in which language and

its purpose, are unique to deaf bilinguals. Consequently, the asset-based/antideficit, deaf bilingualism, and Deaf Gain theories set the direction for the present study.

## 4. Research Question

What are the perspectives and experiences of deaf adult bilingual writers with translanguaging strategies?

### 4.1. Method
Participant Sample and Recruitment Procedures

The sample for this study included fifteen bilingual deaf adults who had previously been identified as high-scoring readers and writers in another study (author 2020) based on several diagnostic exams (Woodcock–Johnson IV passage comprehension subtest, Phoneme Detection Test, handwriting speed test, and Test of Written Language-4). The sample presented the following characteristics: all participants had 85 decibels or more hearing threshold since the age of three or younger, 67% were female, 80% were White, age range 25–53, 87% had ASL as their first language, 60% had two deaf parents, all had family who signed at home, 47% possessed a master's degree, and went to a school for deaf students (in elementary: 53%, in middle school: 67%, and in high school: 60%).

Convenience probability and snowball sampling were used to recruit participants in Nicolarakis's (2020) research. However, for this study, only fifteen bilingual deaf high-scoring readers and writers were chosen from the overall sample to uncover translanguaging strategies that supported their writing process. All participants were recruited or referred through personal and academic networks based on their availability and willingness to participate. Informed consent was obtained from all subjects involved in the study.

### 4.2. Procedure

Each interview lasted approximately 30 min or more, depending on the interview flow and the participants' responses. An interview protocol and guidelines set out by Spradley (1979) kept the questions open-ended and provided opportunities for rich data to emerge. Examples of interview questions were: *What does writing mean to you? Can you walk me through what you just wrote? Can you give me an example of a time you used (if any) ASL to help with writing something? What writing activities did you experience growing up with your family? What are the differences between your writing experience and other deaf individuals? Hearing individuals? I noticed you . . . (insert observation note/question).*

All sessions were videotaped. Afterward, the ASL interviews were translated and typed into English by a team that consisted of the first author, ASL interpreters, fluent ASL/English bilinguals, graduate assistants, and English transcribers to ensure accurate translation (Temple and Young 2004). The ASL interpreters verbally interpreted the video interviews from ASL to English while the English transcribers typed out what the interpreters said in real time. Together, the team read each excerpt and discussed whether the ASL-to-English interview was accurately interpreted, translated, and represented, taking the interviewee's demographics into consideration. If there were any disagreements, then the team deliberated until translations were agreed on unanimously before moving to the next interview. Only twelve interviews were recorded and transcribed from ASL to written English. Two video transcriptions were lost, and one did not record accurately.

Prior to each interview, the first author wrote observation notes from watching the participants' writing behavior during their 30 min spontaneous creative writing assessment using a picture prompt (TOWL-4; Hammill and Larsen 2009) for the Nicolarakis (2020) study. The observations were used to understand the context, translanguaging behaviors, and processes the participants experienced while writing and provided opportunities for follow-up questions during the semistructured interview (Bogdan and Biklen 2007). While the interview aims for this study were not task-based, the role of the spontaneous writing assessment in this study was to provide a more recent recount of the participants' writing

experience and to help them describe their translanguaging writing strategies during the interviews.

*4.3. Researchers' Positionalities*

Onudeah D. Nicolarakis is personally and professionally connected with the present study to demonstrate her commitment to reframing the image of people who are deaf ASL/English bimodal bilinguals (i.e., people who use two modalities and languages to communicate, such as spoken and signed languages) (Abutalebi and Clahsen 2016). As an ASL/English bilingual deaf woman of color, her experiences in each of the intersectional identities have varied. Specifically, she was brought up by a hearing family that spoke English and did not sign. For the first five years of school, she was trained to communicate and learn through spoken and print English only. ASL was forbidden. However, she had a love of writing, which has persisted throughout her life. When she was in school, people complimented her writing by saying she should publish a book or write a blog. This was a source of pride for her, but when she became an adult, she became cognizant of the intimidation and stigma associated with writing for deaf people, who often have negative experiences as writers. As a teacher of the deaf, she saw her students exhibit the same frustration and wanted to teach them to write effectively. It was her way of making equity happen. She also wanted to disconnect the belief that to be a skilled writer, one had to have access to spoken English, because she had had personal interactions with deaf writers who had used ASL from birth.

As a result, Onudeah has both emic (insider) and etic (outsider) perspectives, depending on her role and place within the communities she represents (Reagan 2002). Her academic privilege as a writer because of her access to spoken English and her membership in the deaf communities could be factors in determining whether she obtained the trust of the participants in this study. It was from her interactions and personal beliefs about the deaf communities, along with her intersectional lens, that gave her an advantage in the data collection process for this study. The culturally, linguistically, and academically diverse participant sample from this study trusted her to hold their truths and to ensure their voices were heard. The interviews, while intended to be only for thirty minutes, sometimes lasted for an hour. In addition, the participants connected her with more people in order to help her meet the requirements for the study. The deaf communities had wanted to see her succeed. They knew Onudeah would do her best to represent them well in this study, especially in a field that is about them–but so sorely misrepresented. Conversely, she had caught herself reacting to participants that shared beliefs which countered her own. It was then that the participants were prompted to defend their statements, possibly feeling their truths were being judged by her, potentially affecting the data collection process. Ultimately, it is her goal to keep her eyes open—to question, reflect, and explore with an open mind while at the same time ensuring transparency for those involved in this research.

Thomas Mitchell is a doctoral student in the Critical Studies in the Education of Deaf Learners (CSEDL) program at Gallaudet University, focusing on language, literacy, and curriculum. As a deaf ASL–English bilingual individual, Thomas has had access to signed and written languages since being identified as deaf in the first year of his life. Being the only deaf member in a hearing family that learned and used sign language has left him with firsthand experiences and convictions about the critical role of a natural signed language in the bilingual and bimodal development of deaf children. As a credentialed teacher of the deaf (ToD), Thomas draws from personal and professional experience working with demographically diverse high school students, many of whom are multicultural and bi-/multilingual. He holds a deaf and hard-of-hearing (DHH) teaching credential in California and a certificate for specialized literacy instruction to teach bi-/multilingual learners. The second author's ongoing instructional proximity to translanguaging in the language arts classroom, a common practice in secondary deaf classrooms (Gárate 2012), provides distinctive perspectives on translanguaging as a writing pedagogical tool. As a ToD who works with many deaf writers who are bi-/multilingual and multimodal,

Thomas has often observed the subtractive effects of monolinguistic approaches to writing. Instead, he firmly believes in the liberating and affirming practices in bi-/multilingual and multimodal writing models that find translanguaging at the heart of their orientation (Canagarajah 2011). Thus, emic knowledge, from membership in the deaf communities and as a deaf language arts teacher, has parlayed his acculturation of sociocultural and linguistic values (Reagan 2002) into an innate understanding of translanguaging as a pedagogical tool. Thomas recognizes the duality of his position in this study, both as an intimate participant in the deaf communities and as a deaf researcher who acknowledges that his set of values, experiences, and beliefs are one of many.

## 5. Data Analysis

Theoretical frameworks of asset-based/antideficit, deaf bilingualism/Deaf Gain critical theory (Bauman and Murray 2014; García and Cole 2014; Harper 2010, 2012) guided the selection of the coding process and formation of translanguaging themes. To uncover overarching variables that promoted a rich translanguaging experience in writing success for the high-scoring deaf readers and writers in this study, both causation and elemental coding approaches (Saldaña 2021) were used to code the interview transcripts initially. In the initial coding process, causation coding explored the participants' beliefs about what led to their positive writing outcomes. The purpose was to identify potential antecedental conditions or mediating variables that shaped their writing experiences. Additionally, elemental coding provided the foundation to filter the data for subsequent coding cycles. Structural and in vivo codes formed the basis of elemental coding for data points connected to translanguaging in the first coding cycle. Structural coding positioned translanguaging as a conceptual topic of inquiry that highlighted key concepts, content, and phrases in the interview transcripts. In vivo coding involved the use of the participants' own language and words to develop codes. These first-cycle keywords, phrases, and concepts culled from the transcripts were transferred to a separate list and clustered into focused codes. The authors sorted and organized the focused codes to create thirteen broad categories based on similar characteristics and patterns in the second coding cycle. Subsequently, these categories were then synthesized "toward consolidated meaning" (Saldaña 2021, p. 13) to represent the five thematic categories found in translanguaging in writing by deaf bilingual adults (see Table 1). For example, transcript analyses coded instances of participants discussing how ASL and English influenced each other and the bilingual effect of both languages on their writing outcomes. The first- and second-cycle codes were merged to form the first thematic category of "translingual interdependence". The first- and second-cycle coding process yielded five thematic categories of translanguaging in writing. Translingual interdependence, language flexibility, semiotics and multimodalities, ASL drafting, and visual tracking (see Table 1) emerged through the data analysis coding process (Saldaña 2021), which is discussed in Section 6.

In reviewing 12 interview transcripts as part of the coding process, the authors encountered different perspectives as participants constructed and expressed their meanings of language, culture, and norms in writing. Many interview excerpts resonated with the authors' personal and professional experiences. Still, the authors recognized that importing their values and interpretations into the coding process could compromise the study findings. To reduce the incidence of importing their ideas into the coding process, transcripts were coded based on frequency to identify emergent themes instead of taking the words of participants out of context. The authors frequently conferred to reduce investigator bias in the coding process. Part of this process included dialogue about biases in the coding process and openly questioning the findings and results, no matter how familiar or unfavorable they may seem.

**Table 1.** Thematic categories and focused codes.

| Thematic Categories | Focused Codes |
|---|---|
| **Translingual Interdependence** <br><br> • Views entire linguistic repertoire as valuable writing tool <br> • Encoding <br> • Scaffolding <br> • Bilingual effect <br> • Conscious of language structure and rules <br> • Affects one's self-determination and motivation | • Language influences <br> • ASL as a form of writing empowerment; shed negative label affixed to deaf writers <br> • ASL as a language supports writing and should be included in the writing process <br> • Stages of bilingual writing development (hybrid/fusion of L1 grammar and L2 structure) <br> • Negotiates meaning between two (or more) languages <br> • Use translanguaging to determine or make word meaning <br> • Discusses with other language meaning across languages <br> • Opportunity to accurately represent my language(s) and thoughts across languages |
| **Language Flexibility** <br><br> • Metalinguistic awareness <br> • "Play with language" <br> • Wordplay <br> • "Must know rules to break language" <br> • Alludes to language acquisition device (LAD) theory <br> • Participants often ask "why" behind language use <br> • Tracking visual correctness <br> • Language fluidity | • Abstract, not concrete language use language is a craft <br> • Recognizing and avoiding transposing <br> • English brings form; ASL substance <br> • Language creator as opposed to a language user <br> • Dynamic bi-/multilingual writing as poetry <br> • Decision-making on which language(s) to use for which purpose <br> • Focus on specific areas of writing <br> • Linguistic register <br> • Mental map <br> • ASL English as a disrupter to linguicism/pushing boundaries with language fusion |
| **Semiotics & Multimodalities** <br><br> • How different modalities supported the writing process <br>    ○ Images <br>    ○ Printed text <br>    ○ Videos <br>    ○ Drawings <br>    ○ TV with captions <br>    ○ Lip reading <br>    ○ ASL & nonmanual markers <br>    ○ Comics/graphic novels | ○ Writing books and activities <br> ○ Fingerspelling <br> ○ Gestures <br> ○ Technology (spellcheck, Google, etc.) <br> ○ Signed English markers <br> ○ Facial expressions <br> ○ Written language and its features <br> ○ Signed language and its features <br> ○ Use of ASL space <br> ○ Audio |

**Table 1.** *Cont.*

| Thematic Categories | Focused Codes |
|---|---|
| ASL Drafting | <ul><li>Formal and informal ASL compositions to support written English</li><li>Rehearsing/conceptual development</li><li>transcribing</li><li>Postponing (not very often though)</li><li>Translation and backtranslations</li><li>Editing process</li></ul> | <ul><li>Peer feedback</li><li>Multimodal tools and resources for drafting & writing<ul><li>○ Picture to ASL to English</li><li>○ Drawing and images</li><li>○ Shorthand notes</li><li>○ Mouth morphemes</li></ul></li><li>Fingerspelling</li></ul> |
| Visual Tracking | <ul><li>Visual grammar program</li><li>Fingerspelling to determine correct spelling</li><li>Mouth visemes</li><li>"It looks right"</li></ul> | <ul><li>Visually determining the correctness of writing, including one's own</li><li>Muscle memory (mouthing, signing, fingerspelling, etc.)</li><li>How words "look" on paper</li><li>Rehearsing</li></ul> |

## 6. Results

### 6.1. Deaf Writers and Translanguaging Strategies

A structured coding analysis of interviewees' transcripts demonstrated five thematic areas of translanguaging used by skilled bilingual deaf adult writers: *translingual interdependence*, *language flexibility*, *semiotics and multimodalities*, *ASL drafting*, and *visual tracking skills* (Table 2). Translingual interdependence and language flexibility were translanguaging strategies where deaf writers leveraged their collective linguistic resources to manipulate language and construct meaning in the writing process. Semiotics and multimodalities were tools the deaf writers used to support the planning, drafting, and production of writing. The participants often used the ASL drafting strategy by drafting informal and formal ASL compositions to guide writing in English. Visual tracking as a translanguaging strategy provided deaf writers with nonauditory means to self-regulate what they read or write by determining whether words, phrases, and sentences visually appeared to be correct.

**Table 2.** Five themes of translanguaging in writing by deaf adult bilinguals.

| Five Themes of Translanguaging in Writing by Deaf Adult Bilinguals |
| --- |
| • Translingual interdependence<br>• Language flexibility<br>• Semiotics and multimodalities<br>• ASL drafting<br>• Visual tracking |

### 6.2. Translingual Interdependence

The first strategy, translingual interdependence, occurs when individuals who possess "a plurality of autonomous languages" (García and Wei 2014, p. 11) synergize their collective linguistic system to be greater than the sum of its parts (Zapata and Laman 2016). Participants cited the "bilingual effect" of their available linguistic repertoire as key to understanding content, language scaffolding, and the multidirectional influence of ASL and English on writing. The bilingual deaf adults explicitly discussed how ASL supported their English writing skills in various ways. First, ASL and English supported content mastery by understanding and scaffolding specific concepts in one language (e.g., complex emotions or thoughts) before developing those concepts in a different language. Second, some participants in the study reported that ASL and English provided them with additional means of expressive language skills. Instead of being restricted to one language, participants were able to combine ASL and English language resources for a facilitative effect on their communication and writing skills.

> "This is a hotly debated topic. People often talk about whether ASL is better or English is better but I feel you need both. Each scaffolds the other, ignoring either will be the detriment to both. I firmly believe it's not enough to just master ASL or vice versa. You need both." (DH2)

> "So again, essentially having an understanding of the richness within ASL and all of its nuances really helped me understand similar complexities within English. Making those connections was key for me. Without the ability to understand complex emotions and thoughts in ASL, I would never be able to understand such things in English. You have to have the richness in your native language before you can acquire it in another." (DH3)

> "I think that English is key to navigating through the real world. And you have to emphasize reading and writing equally. You can't let one get ahead of the other. But for deaf and hard of hearing people, it's important to have ASL in there as well. If you focus only on English, it can limit your ability to express yourself.

Having both languages gives you more options to express yourself in different ways." (DH15)

All participants who recognized and discussed one's translingual interdependence viewed ASL and English as interconnected forms of languages that compounded their expressive language options instead of if they only knew one language. More specifically, participants in the study reported that their translingual interdependence was not only essential to their writing experiences but that both languages expanded their capacity for language use and writing.

### 6.3. Language Flexibility

Second, metalinguistic awareness as a part of translingual interdependence was prevalent among the participants. The category of language flexibility demonstrates how the participants viewed the flexibility of language in writing. The ability to be flexible with language represents using one's linguistic register and mental map to leverage bilingualism to impact and manipulate their writing. Participants who referred to language flexibility discussed the ability to play with words and signs to test the boundaries of ASL and English. The participants did not view ASL and English as concrete forms of languages bound by rigid rules. Instead, language flexibility was when the participants traversed both languages, considering how the boundaries of both languages intersect, overlap, and influence one another.

"We have bilingual skills. We know that we can play with words in English. So we want to make sure it's right. ASL is just signed. I noticed at times I would sign something and then repeat with a fingerspelled version of my sign." (DH12)

"Writing itself, it's really a form of expression for me. Sometimes I think better when I'm writing. So it's a way of expressing myself and the way I can sort of play with words." (DH13)

"Words, yes. The more I read and it's not necessarily the craft of writing but the more time I can enjoy playing with words and building the exact phrase I want. I've noticed that when I read less I tend to write just to get my point across." (DH2)

"I enjoy poetry, I like to play with words, and I like experimenting with writing. But I'll get into that another time. I love to test the boundaries of English grammar in my writing, to allow ASL grammar to influence my English. I've done a lot of creative writing like that and even published some." (DH8)

The overlap between translingual interdependence and language flexibility shows that the flexibility in both languages has a multidirectional influence on crafting ASL and English compositions. The participants in the study, when writing, are engaging their collective linguistic repertoire to complete writing tasks while cognizant of how the borders of each language influence each other.

### 6.4. Semiotics and Multimodalities

Semiotics and multimodalities were the third translanguaging strategy that supported the writing process. Participants in the study combined their languages with different semiotics and modalities to broaden the possibilities of their communicative and language intents in ASL and English. Semiotics, or symbolic communication, such as signed English markers, fingerspelling, mouth visemes, facial expressions, and gesturing, are factored into translanguaging as part of the writing process. For example, participants fingerspelled to themselves, a form of multimodality, to make connections between ASL and English. Other examples of semiotics and multimodalities were when participants mouthed the words to themselves to determine the correct word or spelling before writing it on paper. Several participants explicitly noted how fingerspelling and mouth visemes were instrumental to the writing process:

"I think when I sign to myself, well, it's funny. I don't think it out in ASL. I'm more likely to sign to myself. In my mind, I'm thinking about the English structure first, and if I can't figure that out, I'll sign to myself, and that helps me make the connections . . . I'll fingerspell to myself, it probably looks like mumbling, but that's my process for working out how to use both languages to get the information on paper." (DH10)

"So, I'll sign to myself. I either sign to myself or mouth to myself. Why I might, I have no idea . . . But I do . . . I didn't realize I do that. Maybe I'm trying to get it all out there before I forget it. I didn't realize I'm mouthing when I'm writing. I thought it was only when I was reading." (DH13)

"I do mouth some words. I express them that way. And it may have no purpose, but it helps my mental process. Because English, I don't worry about English. It's the way I think, I guess. That's how I'm wired." (DH2)

Additionally, interviewees reported drawing upon multimodalities, such as brainstorming in ASL, writing from images, or making shorthand notes based on ASL and English as various resources to engage their flexible and dynamic linguistic repertoire. These participants noted how graphic organizers, writing prompts, and templates supported their writing process:

"I took a class that introduced all different kinds of strategies for organizing from outlines to spider webs. I saw my motivation improve, my confidence increase, and it gave me the license to play with writing." (DH2)

"I started out by laying out the plot visually. I prefer to use like a line graphic organizer. I can map out the ups and downs of the plot that way. I don't like the graphic organizer that is in quadrants, I feel that's too rigid to put everything in these marked squares. I found drawing the story arc allows fluidity in the kinds of stories you produce." (DH8)

"I need a prompt. I can't start with a blank page. If you tell me where to start, I can branch off. I can expand, but I can't start from nothing. I'm not an initiator with words. With other things, sure, but with writing I need a template, or a script—not a script, but a structure that I can model my writing on." (DH15)

Writing is not unimodal, and this study provided insights into how skilled deaf writers strategically utilized semiotics and multimodalities to support the writing process. Participants in the study recalled times when they used semiotics and multimodalities to complete writing tasks:

"I remember once when I was in middle school, I wrote something, and I was looking for the word 'dark.' I couldn't come up with it, so I just drew a black spot on the page . . . For me, as a user of ASL, it's a visual language. So a black spot clearly meant 'dark.' But the teacher told me I had to use words, I couldn't use symbols....Now with emojis, I use words even less. Emojis can be even more direct than words." (DH3)

Additional resources to support multimodal writing identified in the study include word processing tools such as spellcheck, consulting the website for help, and cross-examining multilingual texts, which are considered multilingual resources (Hesson et al. 2014).

*6.5. ASL Drafting*

ASL drafting emerged as the fourth translanguaging strategy in the study, where participants utilized ASL before translanguaging keywords, ideas, or even full excerpts into English. ASL drafting can be an informal or formal composition that deaf writers use to plan a written piece. Participants who reported different versions of ASL drafting signed their ideas or concepts to themselves prior to transferring this information to paper. Some reasons for signing to themselves were to organize their thoughts and to send forth their best-written work or to draw from a broader repertoire of linguistic resources to strengthen

or clarify their ideas in writing. Variations of translanguaging practices such as rehearsing, transcribing, postponing, translation, and backtranslations occurred in the ASL drafting category. For example, one participant stated:

> "So I learned signing information helps me be able to explain myself better in writing. Since then when I'm giving a presentation or creating a vlog or sending an important email I'll practice in sign language first. Just to make sure that it makes sense before I write it down. For minor matters I don't need to do that but if it's something important or highly sensitive and I want to make sure I give it the respect that it's due when I use that strategy." (DH9)

ASL drafting includes the complex process of translating and transcribing ASL compositions into English compositions, rehearsing, and postponement in these skilled deaf adult writers:

> "But ASL offers me the opportunity to play with the language, to come up with ideas more rapidly than if I were just typing everything out in English which would just be too much. Sometimes what I will do is sign out ideas to myself, videotape myself, and then write them down in English based on the videotape." (DH3)

> "When I am writing something for myself, and I look at it and realize that it is not something I wanted to say, I will sign it to myself and that will help me choose the English word I want. One English word can have so many different meanings and in ASL I have a lot of choices, too. Sometimes signing it to myself in ASL will help me develop the concept and understand what English words I want to use to represent that concept more clearly." (DH4)

> "Signing before I write the information helps me to clarify my intent . . . I tend to have somebody sit down with me to give me feedback and based on their reaction I make adjustments. So, their input helps. If I sign it and get no reaction at all I realized I've missed the mark. So, getting someone else's feedback helps me to make necessary adjustments. I don't do that too often but when I do it really helps." (DH9)

The prewriting of these deaf adult bilingual writers did not adhere to an exact formula. Still, the use of ASL was a prominent feature in developing and translating key concepts and ideas into English written form. ASL drafting also supported the planning component of the writing process when participants used ASL to brainstorm ideas, organization, and develop details.

> "I use ASL to get started with the concept, to come up with ideas, and get something out there. Then I fine-tune it with English. Once I have my general concept in ASL, then I hone in on the details with English." (DH11)

> "So, when I'm creating a story I use ASL first because I'm able to come up with a plethora of specifics that will help me when I later write it in English. I generate more detailed ideas in ASL that I can transfer into written English. After that, I can determine specific word choices while writing in English." (DH2)

### 6.6. Visual Tracking

The fifth and final strategy identified in the study was visual tracking and ASL skills to write, revise, and edit their written works. Iterations in the data showed that participants discussed their innate sense of linguistic rules and structures that help them determine if their writing "looks right". Participants in the study reported using visual and ASL tracking skills to edit and revise their written works. The use of visual tracking allowed the participants to engage in a form of self-correction to check the accuracy of their work. Self-correction through visual tracking had broad applications, such as checking for spelling and grammar accuracy. More saliently, instead of relying on sound, the individuals in

the study used orthographic processes to visually track and determine the correctness of written words or excerpts, including their own:

"You have seen hearing people actually speaking aloud what they are writing. And it looks like it's an auditory feedback loop, where I don't have that. They are stuck into their feedback loop. They can't let go of that. (Interviewer asks DH3 if they have their own feedback loop) It's visual. I don't know. I just visually check what it looks like, literally from the hand up to the eye and back." (DH3)

"Well, yeah because I can't rely on the sound for the punctuation. I don't have the audible cues to rely on for example whether it is a semicolon, comma, or what. In that video you gave me there was a list but I couldn't decipher what punctuation to use. Because I'm not relying on the sound I'm hyperaware of what the punctuation is supposed to look like in a sentence." (DH2)

"If I write something and it looks strange and I need to double check the word, I will fingerspell it to check myself. If I thought of a simple word, but I want to choose a more advanced vocabulary word, I will look at my hand to see if the shape of the word is right and that helps me double-check if the letters in the word I have are written in the right order." (DH4)

"If I'm hesitating, I would write it out and look at it and see if it doesn't look right. And if I'm still hesitating, I would not only write it out but fingerspell it, too. I'll fingerspell it again and again until I feel I've gotten the right spelling. And sometimes it makes me reconsider if I've been fingerspelling it right or wrong this whole time. But I do have the sense right away if it looks funny, if a word looks off." (DH14)

Specific orthographic learning processes used by the participants in the study helped visually track and self-monitor their own writing processes. During this time, the participants visualized the relationship between signed forms or ASL and written forms of English to support or check their written work. Thus, the participants in the study appeared to use forms of orthographic mapping to visually track the correctness of their compositions instead of phonological strategies, such as examining the phoneme–grapheme relationships or word decoding. In other words, the participants based their self-monitoring on whether their compositions "looked" right, not "sounded" right.

### 6.7. Incidental Findings

While this study aimed to uncover translanguaging writing strategies and tools, participants revealed and discussed early experiences that served as antecedents to translanguaging in writing. Recurrent themes in the coding process showed that family involvement and reading and writing connections served as a precursor to bilingual language skills and, subsequently, translanguaging in writing. Family involvement included using ASL at home, positive early reading experiences, and translanguaging modeling. Participants in the study recalled specific instances in their childhood where they engaged in various acts of shuttling back and forth between ASL and English in the home and school. They did not specifically recall these incidents as translanguaging, but they were able to explain scenarios where their parents modeled the use of ASL and English. Furthermore, these participants viewed these interactions as positive early literacy experiences.

"One of the most important things that they emphasized was reading and English and then discussing the explanation in ASL. Then I would explain to them back in ASL and we would reference the writing and we would practice it that way." (DH11)

"When I was growing up my mother was a reading and writing teacher, and she made sure my brother and I were comfortable with writing. My parents really valued literacy in both ASL and English. They made sure we understood the concepts behind what we read." (DH13)

> "I had a positive signing environment in elementary school, good language base, I grew up reading comic books." (DH12)

All but one participant recalled early positive reading experiences and considered themselves "avid readers". Positive early reading experiences included ASL–English read-aloud experiences with parents, caregivers, and school personnel who modeled various principles of reading and translanguaging strategies.

> "The other theory that my mother had was to read books pointing at words while she signed what she was saying, going back and forth between the words on the page and signing. With words like "the" obviously she just had to spell them." (DH9)

> "One of the most important things that they emphasized was reading and English and then discussing the explanation in ASL. Then I would explain to them back in ASL and we would reference the writing and we would practice it that way." (DH11)

Participants connected their reading skills and experiences to writing skills, such as understanding literary devices and features and building language, vocabulary, and grammar.

> "Reading and the constant repetition of seeing proper grammar and punctuation, that's what helps me internalize the rules … Reading is essential for me to internalize punctuation, vocabulary usage and building vocabulary." (DH2)

> "I read a lot of comic books, I enjoyed those. And I think that's where I was able to pick up sort of the social cues, the idioms, dialogue, how people speak. It was easier to understand that stuff better through comics." (DH11)

> "Reading, the older I get, the more I appreciate my parents encouraging me to read. At home we were constantly reading and building my grammar base so I can know what looks right. My mother made sure that we read every night. Now I enjoy reading and see how helpful it is for me. All of that reading helped me develop my innate sense of English grammar so that now I can recognize when something is wrong even if I can't express why or how." (DH13)

However, an emergent theme among interviewees discussed the negative stereotypes of deaf writers and harbored negative feelings or experiences about writing during their school years, lending credence to how the longstanding tradition of monolingual orientation to writing places the mastery of prescriptive English language practices at the center of writing excellence, deepening the marginalization of plurilingual learners (Vallejo and Dooly 2020).

> "People judge English as being more important." (DH13)

> "We thought English had to be "correct", it had to be exactly like hearing people's English … You have to understand, we had all internalized that idea English was better than ASL." (DH8)

> "I had always compared hearing people more favorably to deaf people in their writing and reading skills, and I had to realize, even hearing people may struggle with those skills." (DH10)

Participants in the study discussed how the shift from monolingual writing orientations to an asset-based view of their bilingual skills provided pathways to resistance and resilience through their writing experiences.

> "I started to realize there was a beauty in the way deaf people used English in the 'for-for' and 'see-see', 'do-do', and other such examples. I started to notice there was poetry in written deaf English, and it inspired my own creativity." (DH8)

> "When writing was on my terms, in my voice, my story, that's when I found the inspiration to write." (DH4)

The deaf bilingual adults in the study reported that ASL was not viewed on par with English nor was ASL celebrated or valued in the writing classroom for these deaf individuals. However, through various experiences, the participants noted that incorporating ASL, or any signed language, boosted their views on writing and enhanced their writing experiences. None of the participants reported that ASL harmed their writing experiences but instead said that ASL provided an asset-based reference that positively shaped their views and beliefs about writing.

## 7. Discussion

While findings in the study identified five distinct themes of translanguaging in writing, it was also shown that the interdependence of these strategies was also instrumental to the overall writing practices and experiences of the deaf adult bilingual writers (see Figure 1).

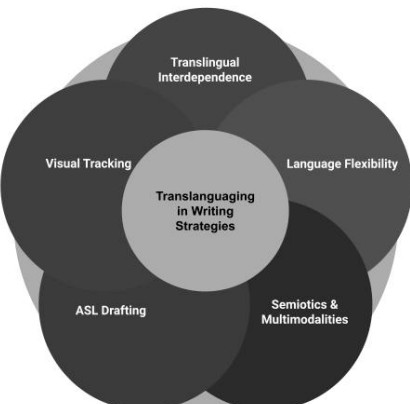

**Figure 1.** Interdependence of translanguaging in writing strategies.

Findings in the study support translanguaging as a naturally occurring phenomenon and an intuitive practice among bi-/multilingual individuals (Canagarajah 2011), including deaf people. Translanguaging practices by deaf adult bilingual writers suggest that these practices are essential to the metacognitive and executive functioning skills needed for writing (Makalela 2015; Velasco and García 2014), as the participants used their combined linguistic and semiotic repertoires to plan, scaffold, and complete their writing tasks. The strategies identified in the study also point to an element of metalinguistic awareness as participants manipulated their entire linguistic system to complete writing tasks, particularly with ASL as the predecessor to written English.

Participants in the study viewed the translingual interdependence of their ASL and English linguistic resources as a critical asset to the writing process. Many participants reported that their families doubled as language resources and translanguaging models who played pivotal roles in developing their language and writing skills. The participants' use of ASL and English, or their collective linguistic and semiotic repertoires, to support and complete writing tasks suggest that monolingual (and subtractive bilingualism) writing orientations in the classroom place deaf writers at a disadvantage. While the participants in the study were strictly bilingual, it is essential to note that research (e.g., García and Wei 2014) shows that bi-/multilingual learners benefit from incorporating all their languages in writing instruction, not just two languages (e.g., ASL and English). The translingual interdependence of languages suggests that deaf multilingual learners (DMLs) may benefit from including a natural signed language, the dominant spoken language, and their home language to scaffold their linguistic resources (Scott et al. 2022). As the participants in the study conveyed, the ability to tap into one's available linguistic resources enriches the language and writing experiences of these bilingual individuals. The benefits of translingual interdependence in the study proffer the premise that the collective linguistic and semiotic repertoires, or all the languages and modalities the students possess, are instrumental to

the writing process. That is, the deaf learner's signed and spoken language(s) may be cooperatively used to facilitate the writing process. The implication is that part of the translanguaging writing approach needs to include strategic and explicit instruction on metalinguistic awareness and how to transverse different linguistic resources to become an accomplished writer.

In addition to translingual interdependence, the deaf bilingual adult writers were cognizant of their metalinguistic awareness and how language is meant to be manipulated for communicative and writing purposes. This finding implies that deaf adult bilingual writers can manipulate signs or words, phrases, and entire compositions in one language and across languages. Emergent deaf bi-/multilinguals will benefit from explicit instruction on the abstract nature of language (e.g., multiple word meanings or manipulating onset and rimes) in addition to how 1:1 word correspondence does not naturally exist between two or more languages (e.g., "C–A–T" may be signed in more than one way). For instance, teachers may want to present an ASL classifier, sign, phrase, or even concept and have deaf students come up with various English translations of the ASL version to help them understand that language is dynamic, flexible, and meant to be manipulated.

Semiotics and multimodalities served as an extensive network of linguistic resources that enabled the deaf adult bilinguals in the study to bypass traditional monolingual orientations to writing to generate unique written compositions. The participants in the study encompassed a variety of semiotics and multimodalities, such as fingerspelling, mouth morphemes, drawings, and more, to scaffold the writing process. Each participant preferred semiotics and multimodalities as part of the writing process, but none approached writing within the strict boundaries of linear English. This infers that there may be applications in the writing classroom as teachers may consider the vast resources of semiotics and multimodalities available to support the writing process. For example, one participant drew a black dot instead of writing the term "dark" to avoid interruptions to the writing process because they did not know the term at the time. Considering this finding, deaf learners might be encouraged to embrace those features of ASL as a form of translanguaging in writing. However, simply presenting semiotics and multimodalities as an option may not be sufficient. It may be necessary for teachers to actively model and encourage the use of ASL features to self-regulate the writing process.

Translanguaging strategies in deaf adult bilingual writers coincide with previous research, such as rehearsing, transcribing, postponing, and translation, to name a few (Gárate 2012). To summarize this category, the term "ASL drafting" was coined to primarily characterize the preplanning and sequential nature of ASL-English writing approaches found in the participants. In the study, ASL was foundational at every turn of the writing process and often preceded and informed the English compositions. The implication is that ASL, even though it lacks a written modality, supports the writing process for bilingual deaf adults in the study and does not hinder or prevent the development of writing skills in these individuals. Therefore, there are two potential applications to the finding that fluent deaf adult bilingual writers use and view translanguaging in ASL as instrumental to the writing process. First, writing instruction in the classroom may benefit from preplanning and drafting processes in ASL before preplanning and drafting in English. Furthermore, translanguaging strategies in writing point to the importance of considering the home language of DMLs in addition to a natural signed language. Furthermore, deaf learners may benefit from a dynamic bilingual classroom where ASL and English are taught as formal courses for credit to capture the bilingual effect in writing. The second implication is that deaf writers may need as much time and as many opportunities to develop and draft ASL compositions as they do with their English compositions. For example, students may prepare a personal narrative in ASL that emphasizes its components (e.g., beginning, middle, end, etc.) and then draft an English version of their narrative with these same elements of narrative writing.

Many linguists studied and advanced the idea that language is an innate human experience and that the brain is hard-wired to detect and incorporate language patterns

and skills, especially during the critical years of language development (Easterbrooks and Baker 2002). Visual tracking represents the deaf participants' innate understanding of language as these participants often referred to visually checking their spelling or grammar to see if it looks right. The bilingual deaf adults in the study used various resources to visually track language, including fingerspelling and visual perception, to determine if the words, phrases, or sentences appeared correct. These findings suggest that orthographic processes may play a significant role in the literacy experiences of deaf bilingual adults in the study. All but one participant referred to themselves as avid or strong readers who felt that their reading experiences were pivotal in their English language development. This suggests that these bilinguals may have relied on orthographic processing of a spoken language, not phonology, and that reading became essential to developing an innate sense of a speech-based language. A potential implication for deaf learners is how reading may precede writing success, and deaf learners need classroom time (and in the home) to read for pleasure. Classroom instruction may stand to benefit from this information in two senses. First, reading for pleasure needs to become a mainstay in the classroom regularly scheduled reading activities (e.g., Drop Everything and Read or Silent Sustained Reading) and engaging reading assignments. Second, teachers' use of modeling or think-aloud may be a promising writing instructional tool (Baker et al. 2003) by cultivating visual tracking skills in emergent bilingual deaf writers. For example, teachers may model or think aloud how to self-regulate the writing process by using fingerspelling, which has been shown to have a positive relationship with overall English literacy development (Alawad and Musyoka 2018).

Another significant finding in the study was the prevalence of monolingual writing orientations in the classroom, which led some participants to internalize negative feelings and experiences about the writing process. Negative stereotypes and feelings of vernacular "Deaf English", "imposter syndrome", and "hearing gaze" were recurrent themes by skilled bilingual deaf adult writers. Despite these limitations, some participants in the study expressed that an inclusive writing experience encompassing both languages was a pathway to empowered writing experiences and sharing their voices as deaf individuals in the world. This supports the reframing of the deaf writer experience to reflect an *asset-based/antideficit*, *deaf bilingualism*, *and Deaf Gain critical theory* that celebrates the bi-/multilingual experiences of these writers (Bauman and Murray 2014; García and Cole 2014; Harper 2010, 2012). Furthermore, to extend the discussion to multiply-marginalized deaf communities, Gárate-Estes et al. (2021) found that Latinx deaf youth used translanguaging to resist forms of linguicism, ableism, and racialized experiences in the deaf classroom. Thus, the intentional creation of translanguaging spaces in the writing curriculum is needed (Escobar 2019; Hesson et al. 2014), and these safe spaces may create opportunities for translanguagers to develop their entire linguistic skills outside of monolingual writing environments (Otheguy et al. 2015, p. 83).

## 8. Limitations

One purpose of the study was to add to the literature on translanguaging strategies in writing by examining deaf adult bilingual writers to be used as a starting point for future intervention studies with PreK-12 deaf learners. Most studies in translanguaging and deaf people are limited to deaf bi-/multilingual adults and focus on the overall language capacities of those individuals. However, these come with some limitations. The data collected in this study relied on the self-reporting of the participants. Another significant limitation of the study was the convenience sample, which did not represent the heterogeneous deaf population. For instance, 80% of the participants were White, 87% of the participant's first language was ASL, approximately 60% had deaf parents, and all participants had access to sign language at home. Another limitation was the potential influence the spontaneous writing assessment had on some of the participants' responses to the interview questions, which possibly limited their reflections to a particular moment rather than a holistic overview of their writing experience. Another limitation was the loss

of interview data from the three participants in our study. However, the benefit of this study's qualitative approach was to provide insight into conditions that promote successful dynamic bilingualism and translanguaging experiences with writing for deaf adults.

## 9. Future Research

The presence of translanguaging in writing strategies in deaf adult bilingual writers is promising. Future research must identify how translanguaging in writing is best incorporated and taught across language and writing curricula for PreK-12 deaf learners to scaffold their writing experiences. More research is needed to determine the strength of the relationship between the antecedents mentioned in the study (e.g., family involvement and reading/writing connections) and translanguaging in writing. Furthermore, translanguaging in writing for deaf learners needs intervention studies that include and elevate the voices of marginalized languages within the deaf communities and discuss how their languages can be a part of the translanguaging writing experience.

## 10. Conclusions

Translanguaging strategies are an essential part of the writing process for deaf bilingual adults. The findings in the study exhibit that those fluent deaf adult bilinguals who are accomplished writers utilize a range of translanguaging in writing strategies to accomplish various writing tasks as they navigate between their ASL and English linguistic and semiotic resources. The study identified five interrelated translanguaging in writing strategies—translingual interdependence, play with words, semiotics and multimodalities, ASL drafting, and visual tracking—that illuminate the power of dynamic bilingualism and can be leveraged into a dynamic writing process. The use of translanguaging strategies demonstrates heightened metacognitive skills and metalinguistic awareness, resulting in language enrichment, content understanding, and more profound writing skills for the deaf bilingual adults in the study. Overall, the findings support that including all languages (e.g., natural signed language, a home language other than English, and the dominant spoken language) in writing instruction for deaf learners is an essential part of translanguaging in writing experience. Therefore, deaf individuals who can use their collective linguistic and semiotic repertoire in writing will have additional resources for a positive and successful writing experience.

**Author Contributions:** Writing—original draft, O.D.N. and T.M. This was a collaborative effort by both authors. All authors have read and agreed to the published version of the manuscript.

**Funding:** This research received no external funding.

**Institutional Review Board Statement:** The study was conducted in accordance with the Declaration of Helsinki, and approved by the Institutional Review Board of Teachers College, Columbia University (19-207, 14 February 2019) for studies involving humans.

**Informed Consent Statement:** Informed consent was obtained from all subjects involved in the study.

**Data Availability Statement:** No new data were created or analyzed in this study. Data sharing is not applicable to this article.

**Conflicts of Interest:** The authors declare no conflict of interest.

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
