# Peer review of "Dynamic Bilingualism to Dynamic Writing: Using Translanguaging Strategies and Tools"

_languages, doi:10.3390/languages8020141_

Round 1

Reviewer 1 Report

Thank you for allowing me an opportunity to engage with your manuscript. The integration of translanguaging with the deaf community is so important, and it is done very nicely here. Below I offer some suggestions. 

Introduction

The introduction provides a solid foundation for readers’ understanding of translanguaging, its role in (multilingual/multimodal) writing, its benefits, and the need to create and sustain intentional pedagogies that leverage it.

Line 111: There is a repeated word here.

Theoretical Framework

Unless you are saying that your theoretical framework is the same as what you have already included in the introduction, this section could benefit from being further developed.

Method

The discussion that both authors provided on emic and etic perspectives was appreciated and helpful in understanding their experiences, biases, and processes as they related to the study.

Line 314: There is an incomplete sentence

Line 329: Given that you are using a translanguaging lens for this study, it is particularly important to use strengths-based terms. Consider using “emergent bilingual,” “bilingual,” or “multilingual” instead of limited English proficient.

The coding table provides an enhanced level of trustworthiness to the study.

Results

The first paragraph in this section provides a helpful roadmap of what is to come, including definitions and explanations.

Line 576: There is a typo here

The statement that there were “precursors” to translanguaging is interesting. From my perspective as a reader, it seems like these precursors are just normal occurrences in bi/multilingual and/or bi/multimodal contexts. What led you to believe they are precursors here any more than being exposed to two or more autonomous languages could lead to bi/multilingualism?

Discussion

A challenge that I have as I read through the discussion is that many of these claims relate to teaching, but the findings were about how adults wrote, not learned in class. Were there more data that were not discussed in the article? At best, you can state something like, “the data suggest that…” or “it may be that…” But, the data do not currently support such authoritative statements.

Line 264: It’s even more, right? Your study shows that the participants regularly drew on their entire multilingual AND multimodal/semiotic repertoires.

Line 652: There is a typo here

Lines 654-5: There seems to be a repetition of words here.

Lines 656-61: If the argument is to first present ASL, consider just saying that instead of starting with the English-first example.

Line 674: Do you mean INclude instead of PREclude?

Limitations: It is important to note that there were no observations in this study; it was comprised of participants’ self-report. Also, if the authors are ultimately interested in PreK-12 intervention, mention should be made that adults (and not PreK-12 students) were the particiapnts here.

Author Response

Thank you for giving us the opportunity to submit a revised draft of our manuscript titled Dynamic Bilingualism to Dynamic Writing: Using Translanguaging Strategies and Tools to Languages. We appreciate the time and effort that the reviewers have dedicated to providing your valuable feedback on our manuscript. We are grateful to the reviewers for their insightful comments on our paper. We have been able to incorporate changes to reflect most of the suggestions provided by the reviewers. We have highlighted the changes within the manuscript as well as submitted a clean copy. See the point-by-point response to the reviewers’ comments and concerns in the attachment.

Reviewer 2 Report

Thank you for sharing your work. First, I would like to recognise the need for more studies on writing as a process in this unique population. I thought the paper presents several important points regarding how translanguaging strategies are instrumental in the writing process in deaf population. However, there are several issues in your paper. 

Introduction/background could be improved in terms of content. There were several points in the introduction that were difficult to follow. 

  • Lines 34/35: what do you mean by 'leverage the entirety of their multilinguistic resources to amplify their linguistic competencies"? If they already have the resources, they are not amplifying their linguistic competencies because they already are competent to begin. 
  • Lines 34/45: I would revisit using the term 'entire.' Maybe instead using entire, use the words like available or collective. 
  • Lines 48/49: What do you mean that use of translanguaging 'can transcend prescriptive language practice and norms'? 
  • Lines 64/65: The first sentence was difficult to follow for several reasons. 1. It talks about translanguaging as a pedagogical tool and then writing process. You could improve the content to make it clearer by providing us some contexts of writing as a process and then how translanguaging strategies have found to support the process.
  • Lines 75-107: I think you would do well to dedicate a paragraph on language ideology/attitude and how it affects writing process particularly those who are bi-/multilingual learners. The topic sentence does not really follow with the rest of the content. In the beginning you offer an argument on the difference between using translanguaging strategies intentionally as opposed to intuitively. Then, the rest of the section talks about monolingualism, the effect of labels, etc. There are several important ideas/points in that section that warrant its own paragraph. 
  • Lines 143: What do you mean by 'leaves researchers with more questions than answers'? Because even if there is a plethora of research, there will always be more questions.
  • Lines 163/164: using the word 'must' seems bold especially when you said there is not enough research. How can we justify that translanguaging theory must remain at the forefront of all.... when we do not have research evidence to support this statement. 
  • Line 170: I am slightly confused because earlier in the paragraph it says there is limited research but here you say a growing number of research. 
  • Lines187-196: What do you mean you want to find out what strategies are relevant to the deaf learner? It would be difficult to identify what strategies are or are not relevant. Also in later paragraph, you describe different strategies in the context of bilingual education in deaf population. 
  • Lines 212-222: Would be useful to review the paragraph to make it more streamlined. Perhaps along the line likes because there is xxxxxx, this study aims to fulfil by....
  • Lines 266-267: It is not clear as to how you control for fidelity. Did you mean that the first author translated and then ASL interpreters, fluent ASL/English etc. ensure accurate translation. How exactly? Did they also watch the videos? Do they evaluate the first author's translation together or separately? 
  • Researchers Positionalities: I think this section could be improved, making it more concise and clear, especially in how you control or check for research biases.  
  • Lines 296-297: The sentence is not clear, also what do you mean by 'she' represents? 
  • Line 327: 'holds special authorization to teach,' did you mean teaching license or teaching certificate? 
  • Lines 352-358: If transcripts were already reviewed by a team like you said of interpreters, ASL/BSL bilinguals, etc... surely this should reduce or ensure reduced/free of bias during the coding process? 
  • Lines 374-376: How do you determine thirteen broad categories? And when I look at the table, it does not seem to align with 13 broad categories. What is the difference between coding categories and focused codes?
  • Lines395-399: You seem to be proposing a model in relation to translanguaging strategies in deaf adult bilingual writers. It is difficult to recognise the model if you have not presented results yet. Perhaps hold onto the model until after you have presented results (data) maybe where you offer interpretation in later paper (discussion/conclusion). 
  • You would do well to review the structure of your results. There were times where it was difficult to follow especially the transition between the author and the data. Would be useful to think about how to add some kind of visual effect to help readers to see data in the results section clearer/better. 
  • The results could be improved to describe more about how those data inform broad categories, 
  • Lines 629-630: What do you mean by 'This view was also shared by the participants' families...' I think you meant that participants explained that their families shared similar view if I am not mistaken. 
  • Lines 641-644: I am not clear as to what you are trying to say. Because we at one point did not have the linguistic repertoire to write but we learn as we write overtime. This brings me to wonder the significance of classroom discourse for pre-writing and during the process. 
  • Lines 695-701: This part is confusing. What do you mean by, '...reading became essential to developing an innate sense of a speech-based language."? Because you were not clear as to who you were talking about. Did you mean bilingual learners in general or did you mean specifically deaf bilingual learners? 
  •  I was surprised not to see any discussion on translanguaging classroom discourse. I would think that it would play a significant part in the writing process. Have you by any chance noticed this, read or thought about this? 
  • Limitations section: I would not end a paper like that and would recommend you to move your limitations section or merge it into discussion section. 
  • Overall, I enjoyed reading the article and it made me think about many things in relation to using translanguaging strategies in teaching and learning, writing in your case. I, however, would have liked to see more evidence and examples of translanguaging from the interviews and what those strategies mean for teaching and learning. Lastly, the paper would benefit from editing.
  •  

Author Response

(The authors gave the same response as above.)

Reviewer 3 Report

Thank you for this paper. I enjoyed reading it and engaging with your ideas. I have a few comments which I outline below –

In the abstract you describe your participants as ‘high-achieving’, but this is not adequately explained. High achieving in what way? On p.6 you clarify slightly that you are looking at deaf people who score highly on tests linked to reading and writing. This doesn’t necessarily mean they are ‘high-achieving’ on other metrics. I’d avoid use of terms like this without adequate contextualisation, because it implies that unless deaf people can read and write well, they will inevitably be ‘low-achievers’, which is not true. You also use the word ‘successful writers’ later on to describe these participants. What does this mean? For me, the first definition of a successful writer that springs to mind is someone who has been published, who writes for a living, etc. That may be a naïve interpretation of what ‘successful’ means, but you don’t actually explicitly state anywhere what you mean by these terms. So more clarity in this sense would be really helpful.

P1. Why do you use the term bi-/multilingualism? I would have thought that multilingualism included bilingualism?

P1. You say that “translanguaging is a form of flexible and dynamic bilingualism” – I don’t think this is an accurate representation of translanguaging. By including the term ‘bilingualism’ you’re restricting the meaning – translanguaging is more about using a wide semiotic repertoire, not simply about two named languages. While you do go on to say this later in the sentence, I would remove “bilingualism” here and replace with something like “languaging” or “language practice” to avoid confusion.

P1. “Humans, unlike the brain…” – this is confusingly worded. I see what you mean, but I had to read it twice to understand. Maybe re-phrase this something along the lines of – “Named languages/distinct language borders are social constructions which are not recognised by the language processing centres of our brains”. That’s not perfect, but I think a sentence using that sort of structure would be clearer.

P2. “Translanguaging is a pedagogical writing tool” – I would say, rather, that it can be used as a pedagogical writing tool.

P2. (and throughout) there’s an assumption that the ‘other’ language in the classroom will always be English in this paper. This won’t always be the case! Make sure you specify what/where you are talking about, so that this doesn’t come across as too English-centric.

P3. L111. Achieved “achieve – typo here?

P3. L123/4. Something that crops up sometimes in this paper is lack of signposting when you write. You write here that “translanguaging pedagogy is available to provide… targeted support…” etc. But the reader is left thinking, “OK, so what are these”? Then you write about “in the planning stages of writing…” – are you now covering the targeted support now that you mentioned above? There’s no clear link here. It would be good to see more explicit links and signposting so that it’s clearer how these two things relate to each other.

P4. L145. You have a : rather than a . at the end of the paragraph

P4. L170. “A growing number of research” – should be a growing amount of research, a growing body or research, or similar.

P5. L200. Should be “is used as much as six times as much in deaf classrooms”, or “up to six times as much in deaf classrooms”

P6. Methods section – again this is where problems with signposting/links come up. L248 you say that the interviews were transcribed, but not by who, or from which language to which language. You either need to explain this here, or explicitly say that you discuss this further below.

Similar in the procedure section – it’s not clear whether this interview was task based. You do mention a ‘spontaneous writing assessment’ – what was this? What did it involve? How long was it? How did the fact that it was an assessment impact on the participant’s responses in the interview? Etc. You need a lot more information here.

You talk about the translation/transcription method by saying that ‘first author, BSL interpreters, fluent ASL/English bilinguals, graduate assistants and English transcribers’ were all involved. How? In what order? Were all of these people involved in every transcript? Did you do back-translation to ensure accuracy? What did you actually do? What did you do if there was disagreement about how to translate something (with this many people involved, disagreements must have occurred?)? Etc. You need a lot more information here.

P7. You talk about ‘the’ deaf community. There are many deaf communities, even within a single nation/locality/city there will be multiple communities, and people will have multiple memberships. I would avoid talking about a monolithic deaf community.

P7. Final few sentences of the second authors positionality statement should probably be moved to the data analysis section – it feels out of place here.

Results section is mostly good – some really interesting quotes here! But you’ve used the same quote twice from DH10. I would remove it from one of the two sections. I understand that that single quote fits both themes, but using the same quote twice so close together in the paper suggests that you have a paucity of data, so a less generous reader would see this as a weakness in your data collection. You have enough data in both sections to only use this quote once, I feel.

PAGE NUMBERS RE-START AFTER THE INSERTED FIGURES

P5. Incidental findings – it would be good to support some of these findings with quotes, particularly as you refer to these findings later on (i.e. p6. “this view was also shared by the participants’ families”) with no actual data to back your statement up.

P6. L652. “def” instead of “deaf”?

P6. L654. “rimes”? – is this a typo?

P7. L681. ASL doesn’t lack a written language, it lacks a written modality.

P7. L686. “DMLs” – what is this? I don’t think you’ve used this acronym in the paper before?

P8. ‘Limitations’ section – this feels really tagged on, to be honest. I don’t think it works here. Would be better to move the info about sampling, interview loss etc. to the methods section (it is already there, you can flag it up here as 'limitations' if you want to). Unless the journal specifically asks for a limitations section in this place, I think most of this info could be better placed elsewhere, and you can finish the paper with your Conclusion.

Overall, this is a good paper, I think. You need to look more carefully at the structure and signposting in places so that it’s easier to follow. You also need to define some of your terms a bit more carefully. Overall though, I enjoyed reading this! Thank you.

Author Response

(The authors gave the same response as above.)

Round 2

Reviewer 2 Report

Overall comments: 

Truly enjoyed reading the paper and can see some improvement. The topic is interesting and important. However, the quality of paper could still benefit from further editing/revising.  

Here are some examples: 

When introducing a new concept or key term, a brief explanation would be useful.  

For example, in the paper,  when talking about plurilingual learners for the first time. There was no background information in relation to where it came from and/or how it is different from multilingual learners that has been used to describe throughout the paper. 

The last sentence of the first paragraph on page 7 could be improved/broken down into sentences. Also, it is not clear when you say translanguaging spaces in the writing classroom need to allow… Did you mean it will allow students? Also, what do you mean by ‘to resist’ if the translingual view is incorporated/practiced in the classroom?  

The last paragraph on p. 8: it appears repetitive. It is already mentioned that there is not enough research in relation to translanguaging as a tool for writing in deaf individuals *see p. 7, then again later on p. 9. 

Tell us more about Horner and Tetreault’s study… is it in relation to deaf or bi-/multilingual learners in general?  

The last sentence before presenting theoretical framework on p.10 is not clear. What do you mean by ‘who are sources of translanguaging strategies in writing.’ 

The data in relation to high-scoring readers and writers need more information. From what another study? What scores did they use?  

Visual representation of the participant sample would be nice addition.  

Only 12 interviews were recorded and transcribed, that sentence should be moved later when discussing procedure.  

The last paragraph re: participant sample section on p. 12 could be improved for clarity. Did you mean that the participants were reached out from another study?  

How accuracy in interpretation is established is not clear. Also, was there any disagreement in translation/transcribing? If so, how was it addressed?  

The third paragraph re: observation does not seem to align with the previous two paragraphs. Were they interviewed and also filmed doing a writing task? This section could be improved for clarity.  

The section on positionalities of the researchers could be improved for conciseness/clarity.  Who is Thomas? See on p. 15 in the second paragraph.  

The data analysis could use some clarification. For example, how are causality and elemental coding approaches different? What are structural and in-vivo codes?  

Perhaps a concrete example or two to help readers understand the process in how you analyze the data would be useful. Especially when consolidating into five different meanings. 

There were no clear descriptions for each category. Perhaps detailed aspect of each category would be useful.  

In the paper, it says the five thematic categories but in the table, they are read as categories. Would do well to review for consistency. Another example of inconsistency in the paper is, play with language or play with words. 

In the table, it says (in-vivo coding) under “Play with language” but the rest do not have that. Also, be sure to review for capitalisation of all categories in the table.  

The sentence in the last paragraph on p. 28 needs to be reviewed for clarity. 

The section incidental findings could benefit from conciseness and to be embedded into your discussion section as it is not the scope of your research questions.  

When offering interpretations in the result or discussion section in relation to deaf using specific strategies, tie them with previous studies/wider literature. 

There are also some issues with the discussion section mostly in relation to introducing new points/concepts that were not discussed earlier. Also, the fluency appears confusing at times. Also the focus should be relevant to the scope of the study, but there are times when discussion appears unfocused.  

What is monolingual writing framework? Don’t recall seeing it being introduced earlier in the paper.  

Drop Everything and Read or Silent Sustained Reading, again what are they?  

The last sentence in the first paragraph on p. 35,  must include and accompany? Based on what evidence?  

Using the word ‘should’ should be used cautiously like how it is used in the first paragraph p. 36.  

It is not clear why the participants not preK-12 learners is a limitation when the research question specifically focuses on deaf adults.  

Would do well to strengthen your conclusion to reiterate your findings and how those respond to the research question... such that translanguaging strategies are important part of the writing process in deaf adults. 

Author Response

Dear Reviewer 2,
Thank you for giving us the opportunity to resubmit a revised draft of our manuscript titled Dynamic Bilingualism to Dynamic Writing: Using Translanguaging Strategies and Tools to Languages. We appreciate the time and effort that you have dedicated to providing your valuable feedback on our manuscript. We are grateful for your insightful comments on our paper. We have been able to incorporate changes to reflect most of the suggestions provided by you. We have highlighted the changes within the manuscript. Attached is a point-by-point response to the reviewers’ comments and concerns.

Much appreciated.
